# Crop calendar optimization for climate change adaptation in yam farming in South-Kivu, eastern D.R. Congo

Jean M. Mondo[1,2,3], Géant B. Chuma[1,2,4]*, Henri M. Matiti[1], Jacques B. Kihye[1], Espoir M. Bagula[1], Katcho Karume[1,2], Charles Kahindo[2,5], Anthony Egeru[6], Jackson-Gilbert M. Majaliwa[6], Paterne A. Agre[7], Patrick A. Adebola[7], Asrat Asfaw[7]

1 Faculty of Agriculture and Environmental Sciences, Université Evangélique en Afrique (UEA), Bukavu, Democratic Republic of Congo, 2 Doctoral School of Agroecology and Climate Sciences, Université Evangélique en Afrique (UEA), Bukavu, Democratic Republic of Congo, 3 Department of Agriculture, Université Officielle de Bukavu (UOB), Bukavu, Democratic Republic of Congo, 4 Laplec-UR SPHERES, Department of Geography, University of Liège, Liège, Belgium, 5 Faculty of Sciences, Université Officielle de Bukavu (UOB), Bukavu, Democratic Republic of Congo, 6 RUFORUM, Makerere University Main Campus, Kampala, Uganda, 7 International Institute of Tropical Agriculture (IITA), Ibadan, Nigeria

* geantchuma@uea.ac.cd, geant.ch@gmail.com

**Data Availability Statement:** All relevant data are within the manuscript and its Supporting Information files.

## Abstract

The traditional crop calendar for yam (*Dioscorea* spp.) in South-Kivu, eastern Democratic Republic of Congo (DRC), is becoming increasingly inadequate given the significant climatic variability observed over the last three decades. This study aimed at: (i) assessing trends in weather data across time and space to ascertain climate change, and (ii) optimizing the yam crop calendar for various South-Kivu agro-ecological zones (AEZs) to adapt to the changing climate. The 1990–2022 weather data series were downloaded from the NASA-MERRA platform, bias correction was carried out using local weather stations' records, and analyses were performed using RClimDex 1.9. Local knowledge and CROPWAT 8.0 were used to define planting dates for yam in different AEZs. Results showed the existence of four AEZs in the South-Kivu province, with contrasting altitudes, temperatures, and rainfall patterns. Climate change is real in all these South-Kivu's AEZs, resulting either in rainfall deficits in some areas, or extreme rainfall events in others, with significant temperature increases across all AEZs. Suitable yam planting dates varied with AEZs, September 15th and 20th were recommended for the AEZ 2 while October 15th was optimal for AEZ 1, AEZ 3, and AEZ 4. However, none of the planting date scenarios could meet the yam water requirements in AEZ1, AEZ3, and AEZ4, since the effective rainfall (Pmm) was always inferior to the plant water demand (ETc), meaning that soil water conservation practices are needed for optimum plant growth and yield in these AEZs. This study does not recommend planting yam during the short rainy season owing to prolonged droughts coinciding with critical growth phases of yam, unless supplemental irrigation is envisaged. This study provided insights on the nature of climate change across the past three decades and suggested a yam crop calendar that suits the changing climate of eastern DRC.

**Funding:** Bill & Melinda Gates Foundation (BMGF) through the RTB Breeding project of the International Institute of Tropical Agriculture (IITA) (INV-003446); Carnegie Cooperation of New York through the Regional University Forum for Capacity Building in Agriculture (RUFORUM), grant No RU/2024/Post-Doc/02. The funders had no role in study design, data collection and analysis, decision to publish, or preparation of the manuscript.

**Competing interests:** The authors have declared that no competing interests exist.

# Introduction

Climate change is a serious threat to global food security and poverty alleviation worldwide [1, 2]. Based on several reports, Africa is the most vulnerable continent to climate change owing to overreliance of its rural populations on rainfed agriculture, low adaptive capacity, and insufficient investment in mitigation and resilience-building systems [3–5]. This situation calls for sustainable and climate-smart practices that could strengthen smallholder farmers' adaptation capacity to climate shocks, of which an adapted crop calendar is paramount [6]. A crop calendar helps farmers optimize their yields, minimize the risks associated with weather and pests, and effectively plan their operations throughout the year [7–9]. It considers various factors such as climatic conditions, crop growing seasons, crop life cycles, farming practices, water and labor availability, as well as local constraints, and market requirements [10, 11]. The crop calendar, therefore, includes information on sowing, transplanting, crop management (weed and disease management, fertilization, irrigation, and crop rotation), and harvesting and post-harvest activities [12].

The Democratic Republic of Congo (DRC) has the highest agricultural potential in Africa owing to its conducive climates and fertile soils (i.e., 80 million ha of arable lands), with potential to feed over 2 billion people through suitable investments [13]. More than 70% of its population relies on agriculture for livelihoods [4, 14]. However, rainfed agriculture that is predominant in DRC is highly vulnerable to climate change, especially among small-scale farmers, characterized by limited adaptation capacity [4, 15]. Like other countries in sub-Saharan Africa, DRC is seeking for effective alternatives for countering adverse effects of climate change that are undermining the livelihoods of local communities [4].

In South-Kivu, a post-conflict region, changes in precipitation patterns, increased frequency of extreme weather events, shifting agricultural zones, and cropping calendar threaten the region's socio-economic recovery as they lead to low farm productivity [4]. The situation is further complicated by the lack of effective weather monitoring and prediction system, and non-functional extension system. Consequently, farmers in the region are facing unprecedented challenges associated with climate change, including uncertainties over the optimum planting period [15]. This is most alarming for small-scale producers who depend on traditional crop calendars, based solely on indigenous knowledge, and that are becoming obsolete and ineffective with climate change [6, 16, 17]. Yet, the absence of proper planning increases farmers' vulnerability and leads to significant crop losses, compromising local food security.

Wang et al. [8] showed that adjusting crop calendars may present an effective adaptation measure to avoid crop yield loss and reduce water use in a changing climate. Furthermore, without a reliable crop calendar, it is difficult to timely supply farm inputs or exchange plant material among regions, a necessity for crop genetic improvement. Modern methods for predicting crop calendars, based on weather data and climate forecasts, provide useful information on the timing of certain agricultural activities. These methods use information on climate (mainly crop's water requirement and environmental demand), crop conditions, and soil characteristics [10, 11, 18]. Rainfall characteristics, i.e., rain onset and cessation dates, duration and cumulative rainfall, have important implications for agricultural production, especially in the eastern DRC, where agriculture is essentially rainfed [4, 15]. As defined by Apriyana et al. [19], the onset of the rainy season corresponds to the period at the beginning of the rainy season, when rainfall distribution has become adequate for crop development, while the withdrawal corresponds to the period at the end of the rainy season, when rainfall distribution no longer allows crop growth. Knowledge of such information has led to the development of several tools for assessing crop water requirements, including AQUACROP, Erosion-Productivity Impact Calculator (EPIC), CROPWAT, Decision Support System for Agrotechnology Transfer

(DSSAT), Wine Grape Water Use Model (WINETRO), Crop Estimation through Resource and Environment Synthesis (CERES)-Maize, and others [20]. Of these, CROPWAT developed by the Food and Agriculture Organization of the United Nations (FAO) is one of the most widely used when simulating crop calendars, as opposed to AquaCrop which is mainly used to assess the productivity of crops such as rice, maize, sorghum, etc. [21, 22]. CROPWAT is a user-friendly software package designed to help farmers, irrigation experts, and agricultural planners estimate crop water requirements in different regions of the world [23–25]. It has previously been used to adjust crop and farm input calendars for staple crops such as maize, soybean, and common bean in South-Kivu [26]. This research being complementary to previous efforts in the region, we selected CROPWAT to facilitate comparison of outcomes.

Yam (*Dioscorea* spp.) is a staple food crop playing a central role in many African communities' diets, and thus significantly contributing to food security and sovereignty in Africa owing to its starchy tubers and its ability to adapt to a variety of cropping conditions [27–29]. Beyond its nutritional role, yam supports small-scale farmers' livelihoods, ensures food diversification, and holds a symbolic cultural importance in many communities, including those of eastern DRC [29]. Its resilience to harsh climatic conditions and its use in sustainable farming practices make it an essential crop for subsistence, climate resilience, and heritage preservation [28, 30]. Like other root and tuber crops, yam importance for food sovereignty in Central Africa, and DRC particularly, is expected to increase under the present and future climate change scenarios owing to its high climate change resilience, as compared to cereals and legumes [29, 31, 32].

Limited empirical data exists on the optimum time for different farming activities in yam cultivation in South-Kivu and in DRC in general. Yam farmers still rely on traditional knowledge to decide on the planting date, making their calendar obsolete in the era of climate change. Consequently, Mondo et al. [29] reported a multitude of yam planting and harvesting dates in South-Kivu, implying that there is no consensus on the crop calendar among the region's yam producers, each farmer using his/her own instinct to decide which crop calendar to follow. Since yam cultivation is rainfed, local yam farmers monitor the rainfall onset of the long rainy season, occurring from late August to late October, for planting. Planting early September to late October implies that field preparation starts in dry season (June to August) and that harvesting could be expected from late March to August the next year since yam crop cycle spans from 8 to 12 months. Such a situation (lack of consensual calendar) could make external interventions by actors supporting yam producers difficult, as without a viable crop calendar, the provision of farm inputs such as seed, fertilizers, and other inputs are difficult to plan [26]. Though planting date shifts are effective to avoid yield loss induced by climate change in most crops [8], previous efforts aimed at optimizing the crop calendar in the mountainous South-Kivu did not include yam among target crops [26], probably because of its status as a neglected and underutilized crop in DRC [29, 33]. It is noteworthy that the crop calendar developed for other staple crops cannot be extrapolated to yam since they have different growth cycles and requirements in terms of climatic parameters. The present study was, therefore, carried out to establish a crop calendar for yam cultivation in South-Kivu, using CROPWAT software, to ensure sustainable farming practices, mitigate climatic risks, and facilitate activities' planning by farmer support structures. Specifically, this study aimed at: (i) assessing trends in weather data across time and space to ascertain climate change, and (ii) optimizing the yam cropping calendar for various South-Kivu agro-ecological zones (AEZs) to fit current climate realities. The study area being diverse in terms of soils, climates, and agrosystems, we hypothesize that resulting conclusions will inspire scientists and decision-makers across the continent and the world to develop yam crop calendars suiting climate change in their respective agro-ecologies.

## Materials and methods

### Study area

This study was conducted in South-Kivu province; eastern DRC. This province covers a surface area of ~65,070 km$^2$ and shares borders with Rwanda, Burundi, and Tanzania to the east, North-Kivu province to the North, Maniema to the west, and Tanganyika province to the south. Its population is estimated at 9 million, distributed across eight administrative territories and three cities, of which Bukavu is the largest city. South-Kivu's climate is broadly classified as tropical, characterized by moderate temperatures due to its high-altitudes [29]. Temperatures generally range from 12 to 28˚C, with seasonal variations more pronounced in mountainous areas. Precipitations are high (1500 ± 650 mm), sustaining its forest ecosystems rich in biodiversity [29, 34, 35]. The province also features a diversity of soil types, Ferralsols, Umbrisols, and Acrisols being the most predominant [29]. These varying climate and soil types significantly shape land suitability among staple and perennial crops, including yam. Overall, South-Kivu soils are suitable for agriculture, though soils in some areas require specific management practices such as soil erosion control measures and integrated soil fertility management [29, 36–38].

Agriculture is the main economic activity in South-Kivu, providing employment to >70% of its active population. Main root and tuber crops are cassava, sweetpotato, taro, and yam; common bean and soybean dominate legume crops while maize, sorghum, and rice top cereal crops. Industrial crops are dominated by coffee, tea, sugar cane, and oil palm [39]. However, the state's subsidies are limited, explaining low use of farm inputs among smallholder farmers who are still practicing extensive agriculture [4]. In addition to low governmental investments, South-Kivu agriculture faces unprecedented climate change challenges that threaten efforts in alleviating hunger and poverty among small-scale farmers [35, 37]. A previous study showed that South-Kivu could be subdivided into four AEZs using topographical, climatic, and soil information [29]. These have varying levels of suitability for yam cultivation.

## Methods

### Climate data sources

Analysis of data from existing weather stations enabled to assess the trend of main climatic parameters across the South-Kivu province, with focus on rainfall (daily, monthly, and annual rainfall) and temperature (maximum, minimum, and mean temperatures). In areas where no weather stations existed, open access data was downloaded from the NASA-MERRA platform (https://power.larc.nasa.gov/data-access-viewer/), then edited, and corrected using data from nearby weather stations. Data from weather stations such as CRSN/Lwiro, INERA/Mulungu, the Centre de Recherche en Hydrobiologie (CRH)–Uvira, and the Burundi State Department of Meteorology located in Mparambo, in the Ruzizi Plain, were used to correct online data. We used both data gap filling for daily climate data and the Double Mass Curve Analysis (DMCA) derived from the arithmetic mean, multiple linear regression, and the non–linear iterative partial least–squares algorithm for precipitation data [15]. For the temperature, exponential equations and the non–linear iterative partial least–squares algorithm were used.

### Determination of the nature of weather data's change across time and space

For each AEZ, an umbrothermal diagram was established and interpreted with reference to the following standards: P≤2T (dry month) and P≤4T (very dry month). Similarly, the standardized precipitation index (SPI) was calculated to cluster months and years as dry or wet

based on rainfall patterns. An in-depth assessment of climate risks associated with different planting periods was performed. This assessment implied analyzing rainfall, temperatures, and other climatic factors bearing influence on successful yam cultivation. The 1990–2022 time series' data were analyzed using RClimDex 1.9 package from RStudio [40] to assess climate data trends, and thus, elucidate the nature of climate changes across time. RClimDex 1.9 package is widely used to calculate various climate indices based on daily, monthly, or annual data to identify any significant trends [40]. Specifically, five essential elements were sought for: (i) identification of changes over the last three decades (i.e., determining whether significant changes occurred in climatic parameters such as maximum and minimum temperatures, rainfall, etc.); (ii) characterization of trends (i.e., detect direction (in terms of increase or decrease) and magnitude of observed trends); (iii) detection of seasonal patterns (i.e. identifying specific seasonal trends across time); (iv) variability assessment by examining climate data inter-annual variability to detect stability or instability in climate conditions over the years; and finally, (v) correlation with other phenomena (i.e., exploring the possible correlation between observed climate trends and other phenomena, such as biophysical changes or human activities).

## Procedures used for crop calendar optimization

*Introduction to CROPWAT 8.0 and CLIMWAT.* Several approaches and tools have been developed for decision-making in agriculture. These tools are used by farmers, agronomists, and agricultural planners to optimize crop management based on local climate conditions and soil and crop characteristics [41, 42]. There is no universal tool for decision-making in agriculture, as the choice often depends on the user's specific needs, the user local region's characteristics, and the resource availability. Nevertheless, CROPWAT is the most popular given its convenience [10, 11, 43]. It is widely used to estimate crop water requirements as part of irrigation planning and to model crop growth as a function of climatic conditions. The latest CROPWAT 8.0 version (https://www.fao.org/land-water/databases-and-software/cropwat/en/) was used for simulations, considering South-Kivu climate specificities and local yam varieties' characteristics. Based on findings by Mushagalusa et al. [26], CROPWAT is effective in modeling crop calendars in South-Kivu and was, therefore, used in this study for results' comparison.

CLIMWAT is a tool used to generate weather data for input into the CROPWAT model. It is a weather data generation program developed by the FAO specifically for use with the CROPWAT model. It provides a means to generate historical weather data or climate scenarios required by CROPWAT for simulating crop water requirements, irrigation scheduling, and water balance calculations [42, 43]. It uses historical weather data to generate daily weather variables such as temperature, rainfall, humidity, wind speed, and solar radiation. These generated weather data are then used as input for CROPWAT to simulate crop water requirements and related parameters.

*Model parameterization.* Effective rainfall was determined using the method proposed by the United States Department of Agriculture (USDA) Soil Conservation Service, as per the following formulas [44, 45]:

$$P_{eff} = [(P \times 125 - 0.2 \times 3 \times P)] \; when \; P \leq \frac{250}{3} \, mm \tag{1}$$

$$P_{eff} = \left( \frac{125}{3} + 0.1 \times P \right) \; when \; P > \frac{250}{3} \, mm$$

*With $P_{eff}$: the effective precipitation/rainfall (in mm), P: the monthly precipitation (in mm), in bold are the correction factors used by CROPWAT to adjust the formula in the case of decennial*

*and daily rainfall data. For effective precipitation calculations, daily data are aggregated by decade.*

The method (formula) proposed by Penman-Monteith [46] was integrated into CROP-WAT to estimate the environmental demand that corresponds to the potential evapotranspiration (ETo):

$$ET_0 = \frac{0.408\Delta(R_n - G) + \gamma \frac{900}{T+273} u_2(e_s - e_a)}{\Delta + \gamma(1 + 0.34u_2)} \tag{2}$$

*With ET$_0$: reference evapotranspiration, Rn: net radiation from reference surface (MJm$^{-2}$j$^{-1}$), G: heat flux into soil (MJm$^{-2}$j$^{-1}$), T: daily mean temperature at 2 m (°C), u$_2$: wind speed at 2 m (m/s), e$_s$: saturation vapor pressure (kPa), e$_a$: actual vapor pressure (kPa), e$_s$—e$_a$: saturation vapor pressure deficit (kPa), Δ: slope of saturation vapor pressure curve (kPa/°C), γ: psychrometric constant (kPA/°C).*

Information was integrated into CROPWAT based on yam crop specificities and local farming practices. Five planting dates were tested, based on farmers' opinions [29]. Since the yam life cycle spans from 8 to 12 months (depending on the species and varieties) in the study area, an intermediate life cycle of 10 to 11 months (i.e. 300 to 330 days) was adopted for simulations. The yam crop coefficients (Kc) were sourced from the existing literature [47, 48]. The cycle was then subdivided into five growth stages as presented in **S1 Fig**. The standard plant height was set at 250 cm, and the root length ranged from 10 cm at the initial growth phase to 100 cm in the final growth phase. Since there is limited data on yam, we adapted encoded data for potato, coupled with some locally available data. Therefore, the yield response factor was maintained at 0.45 while the critical depletion fraction was set at 0.25 (**S1 Fig**).

The planting season in tropical areas is generally determined by the rainfall onset, as water is the main limiting factor for agriculture [49], though it can also be decided based on local farming practices [50, 51]. Therefore, it is obvious that, for the same crop under the same climatological conditions, different planting dates are chosen. This is useful for studying different farming practices and calculating system water supply schedules [41, 42, 52].

The harvest date is automatically calculated based on the planting date and the crop cycle. Yam cropping cycle is subdivided into four stages: (i) *initial phase* extending from planting date to ~10% soil cover, (ii) *development phase* extending from 10% soil cover to effective full soil cover. Effective full cover for many crops occurs at the flowering initiation [53]. (iii) *Mid-season phase* extends from effective full soil coverage to the maturity onset. The onset of maturity is often indicated by the onset of aging, leaf yellowing or senescence, leaf drop, or fruit browning to the point where crop evapotranspiration is reduced relative to reference evapotranspiration. (iv) *The end-of-season phase* extends from the onset of maturity to harvest or full senescence [54–56]. At this stage, ETc calculation is assumed to end when the crop is harvested, dries out naturally, reaches full senescence, or undergoes leaf fall. CROPWAT 8.0 includes data for several common crops from several FAO publications (see Irrigation and Drainage Series No. 56 "Crop Evapotranspiration" and No. 33 "Yield Response to Water". These links require an Internet connection). However, the most reliable crop data are those obtained from local agricultural research stations [57, 58].

Soil: Parameterization was based on soil physicochemical properties in each AEZ. The soil characteristics considered in CROPWAT 8.0 (**Table 1** and **S2 Fig**) were sourced from the Harmonized World Soil Database (HSWD) [59] and Soil, Plant, Atmosphere, and Water (SPAW) 6.02 [60]. HWSD provided information on soil units, total soil moisture, soil texture, and soil density. By selecting this information, SPAN developed by USDA was used to find the values

**Table 1. Soil physicochemical propeeerties integrated into the CROPWAT tool while developing the yam crop calendar for South-Kivu AEZs.**

| Zones | Soil type | Total available soil moisture (TAW) (TAW = FC-WP) (mm/m) | Maximum rain infiltration rate (mm/day) | Maximum rooting depth (cm) | Initial soil moisture depletion (as %TAM) | Initial available soil moisture (IASM, mm/m) |
|---|---|---|---|---|---|---|
| AEZ 1 | *Red Sandy Loam* | 200 | 30 | 300 | 50 | 100 |
| AEZ 2 | *Red Clay* | 200 | 30 | 300 | 50 | 100 |
| AEZ 3 | *Loamy clay* | 220 | 60 | 200 | 50 | 120 |
| AEZ 4 | *Black clay* | 180 | 40 | 100 | 10 | 162 |

AEZ: Agro-ecological zone. Total Available Water (TAW) represents the total amount of water available to the crop. It is defined as the difference in soil moisture content between Field Capacity (FC) and Wilting Point (WP).

for the other columns in the e. **S2 Fig** shows an imaginary volume unit of the root zone with all the elements of the water balance [21].

*Simulation of the planting dates*. In-depth analyses were carried out to understand variations in water deficit across the crop cycle depending on the planting date. The Crop Water Use Calculator is a practical tool designed to help farmers and agricultural scientists and planners determine the crop's daily water requirements [21]. This calculator uses reference evapotranspiration (ETo) and crop factor (Kc) to calculate crop water consumption (ETc) in millimeters per day. This tool simplifies the calculation process and provides fast and accurate results [50, 51]. At the end, the data were provided in terms of decades.

Three output parameters were used to propose the optimum crop calendar: the water demand of the area (in the form of ETc), the actual rainfall, and the difference between the two that equals to the irrigation water demand. These three parameters were provided for each developmental stage, by month of the year subdivided into decades. The optimum crop calendar would correspond to that which minimizes the demand for irrigation water during plant critical phases, particularly the initial/crop establishment phase and the active plant development (corresponding to tuber initiation and bulking), while presenting reduced rainfall/ soil moisture content at maturity to prevent tuber rotting. Scenarios minimizing the water deficit during the first two stages (sprouting and crop establishment) were to be preferred, as yam, like any other root and tuber crop, is too sensitive to water stress during the first months of cultivation [28, 30]. Periods when actual rainfall is less than or equal to ETc correspond to critical phases when supplemental water may be required. In this way, the time intervals proposed by the crop calendar were divided into periods ranging from "less favorable" to "very favorable" for each proposed farming activity. Four main activities were included in the yam crop calendar: (i) field opening and preparation, (ii) planting, (iii) management (including gap filling, weeding, staking, soil fertilization, pest control, etc.), and (iv) harvesting. As yam varieties used for analyses are late maturing spanning from 10 to 12 months, only a single crop is possible yearly, meaning that cropping activities of yam crop calendar extend across the two cropping seasons (long rainy season A and short rainy season B) that characterizes the South-Kivu.

To validate optimum scenarios and planting and harvesting dates across AEZs, the water requirement curve (*ETc*) was compared with the effective rainfall curve (Effective rain) for each dekad of the month of the year. The dekad in which effective rain exceeded *ETc* was considered as the optimum planting date. If this included a test planting date from our suggested scenarios (or was close to one), the test scenario was validated as optimum. However, time interval of two days (before and/or after) were taken as a margin to the proposed planting

date. Periods when *ETc* is greater than effective rain corresponded to the period of crop management, maturity, or harvest. These periods may correspond to weeding, irrigation, nutrient supply, etc.

### Ethical clearance/statement

The study protocol was approved by the Interdisciplinary Centre of Ethical Research (CIRE) of the Université Evangélique en Afrique (UEA), Ref: CNES 027/DPSK/322PP/2023. We obtained consent from all resource-persons and farmers prior focus group discussions (to select test planting date scenarios) after ensuring the participants of the confidentiality in use of data collected and explaining the study objectives, as approved and directed by the above Institutional Review Board.

### Data analyses

Data analysis was performed using Microsoft Excel, R Studio, and R 4.2.1 [61]. Data processing started with description statistics, i.e., calculations of means, standard deviation, standard error, and coefficient of variation. In addition, the maximum, minimum, annual, monthly, and overall averages of the climatic data were determined. The Mann-Kendall test was used to assess trends in weather data. This test is a non-parametric method used to detect trends in a time series climate data such as precipitation, temperatures, etc. [62]. It assesses whether the data series shows a significant upward or downward trend over time. For climate indices (**S1 Table**), the RClimDex 1.9 package [40] was used for calculations. This was useful to detect significant climate changes over the selected period. For the simulation of water demand, analysis of variance (ANOVA) was used to compare values of water demands among crop growth phases and test planting date scenarios to detect where there are significant differences among them. All these analyses were conducted with a probability threshold set at 5%.

## Results

### Description of the South-Kivu agro-ecological zones (AEZs)

The AEZ 1 characterizes low-altitude zone with an *Aw* type semi-arid tropical climate according to the Koppen-Geiger classification. It experiences average temperatures of ~19°C, with an annual rainfall ranging between 800 and 1000 mm (**S3A Fig**). With strong variations observed from May to September, it is quite difficult to precisely delimit the dry and wet seasons. May and October are either wet or dry, while June, July, August, and September are generally dry. With <10 mm rainfall, July is the driest month of the year, while November, March, and April are the wettest, recording >150 mm of rain. A difference of nearly 145 to 150 mm of rainfall are observed between the driest and wettest months (**S3A Fig**). Some months, such as May, September, and October show greater variations in precipitation (mm) than others, making farming risky. Areas characterized by the AEZ1 are mainly located in the Ruzizi plain of the Uvira territory, in the north-eastern and southern Fizi, and in the southern Walungu territory towards Kamanyola.

The AEZ 2 is characterized by low-altitude forest zone, with a sub-equatorial or transitional equatorial climate experienced in the south-western and southern Fizi territory, the north-western Kabare territory, and in the western territories (Shabunda and Mwenga). In this area, rainfall is abundant and distributed throughout the year. This is an *Af*-type climate, with high temperatures (~24°C), and annual rainfall ranging between 2,300 and 2,400 mm. Only June and July have low rainfall, but they still record ~50 mm (**S3B Fig**). Compared to the AEZ 1,

monthly rainfalls are significantly higher, reaching up to 350 mm. More than seven months of the year record rainfall >200 mm, while two have >100 mm.

The AEZ 3 is characteristic of the *Aw3* type humid tropical climate, or mountainous climate with long wintering periods. This AEZ experiences rainfall averaging 1637 ± 21.2 mm annually. Only three months can be classified as dry (June, July, and August) with less than 25 mm of rainfall. These are climatic conditions found in Kabare, Walungu, and Kalehe territories and around the Bukavu city. Four months are very rainy (November, March, April, and December) and three are moderately rainy (May, September, and November). These months also show wide rainfall variations (i.e., shifting from dry to wet depending on the year) (**S3C Fig**).

The AEZ 4 is a humid tropical zone with very high altitudes (>2700 mm), characterized by a temperate climate (17.3 ± 0.7˚C), owing to the high altitude. Indeed, according to Gensler & Buchot [63], for every 190 m rise in altitude, the temperature falls by 1˚C. This high altitude also impacts these areas' rainfall quantity and quality. Lowland areas receive more abundant rainfall, while mountainous regions may experience variable rainfalls. In South-Kivu, areas characterized by the AEZ 4 are located along the Mitumba mountain chain (**S3C Fig**). This climate is diversified due to varying relief. Overall, the region may be subject to a tropical climate, but variations in altitude can create nuances. In lowland areas, a humid tropical climate may prevail, while at higher altitudes, the climate could be cooler (with occasional frosts). Annual rainfall averages 1566 ± 40.2 mm.

## Trends in climate data over the last three decades

The 1990–2022 data series were analyzed, using the RClimDex 1.9 tool, to better understand trends in South-Kivu climate data over time and its potential impact on the proposed yam crop calendar. In the AEZ1 characterizing low-altitude zone with semi-arid tropical climate, there is a slight decrease in moderate rainfall (<10 mm), while heavy rains (>20 mm and 25 mm) increased significantly over the last three decades, leading to recurrent flooding, fields' destruction by siltation from runoff and sediment deposits, plant heaving, etc. Given the curve trend, these extreme rains are likely to increase in the future, causing threats to vulnerable populations' subsistence means. Rains lasting more than a day are infrequent, whereas single-day rains are increasing (**Fig 1**). Precipitation and temperature increased by 12.2% and 5.9%, respectively, from 1990 to 2022. There is 15.6% ($p = 0.022$) increase in rainfall events spanning over 5 successive rainy days, with slight increases in maximum (26.5 to 27.2˚C) and minimum temperatures (14.2 to 14.8˚C) during the study period.

Although the average annual rainfall seems to increase, there is a reduction in the amount of rainfall in wet months. The SPI calculation also predicts more dry years (SPI≤-1.5) than wet years (SPI>1.5) (**S4 Fig**). Looking at the 1990–2022 climate data, series of four to six dry years are often followed by two to three wet years. On the other hand, on a monthly scale, seven months are wet, while the remaining (May to September) are dry. Therefore, if the crop calendar starts with the actual rains' onset, it would be advisable to wait until late October or early November before planting, to escape high variability in rainfall as experienced in October (**Fig 1**).

In the AEZ 2, characteristic of low altitudes with a sub-equatorial climate (also known as transitional equatorial climate), rainfall amounts increased significantly, reaching 2,500 mm yearly. Over time, there is a significant increase for both moderate (<10 mm) and heavy rains (> 20–25 mm). Rainfall events lasting more than 5 successive days also increased in the 1990–2022 climate data series (**Fig 2**). Temperatures also increased significantly, from 27.8 to 29.8˚C, 17.2 to 17.8˚C, and 22.2 to 24.2˚C for maximum, minimum, and mean temperatures,

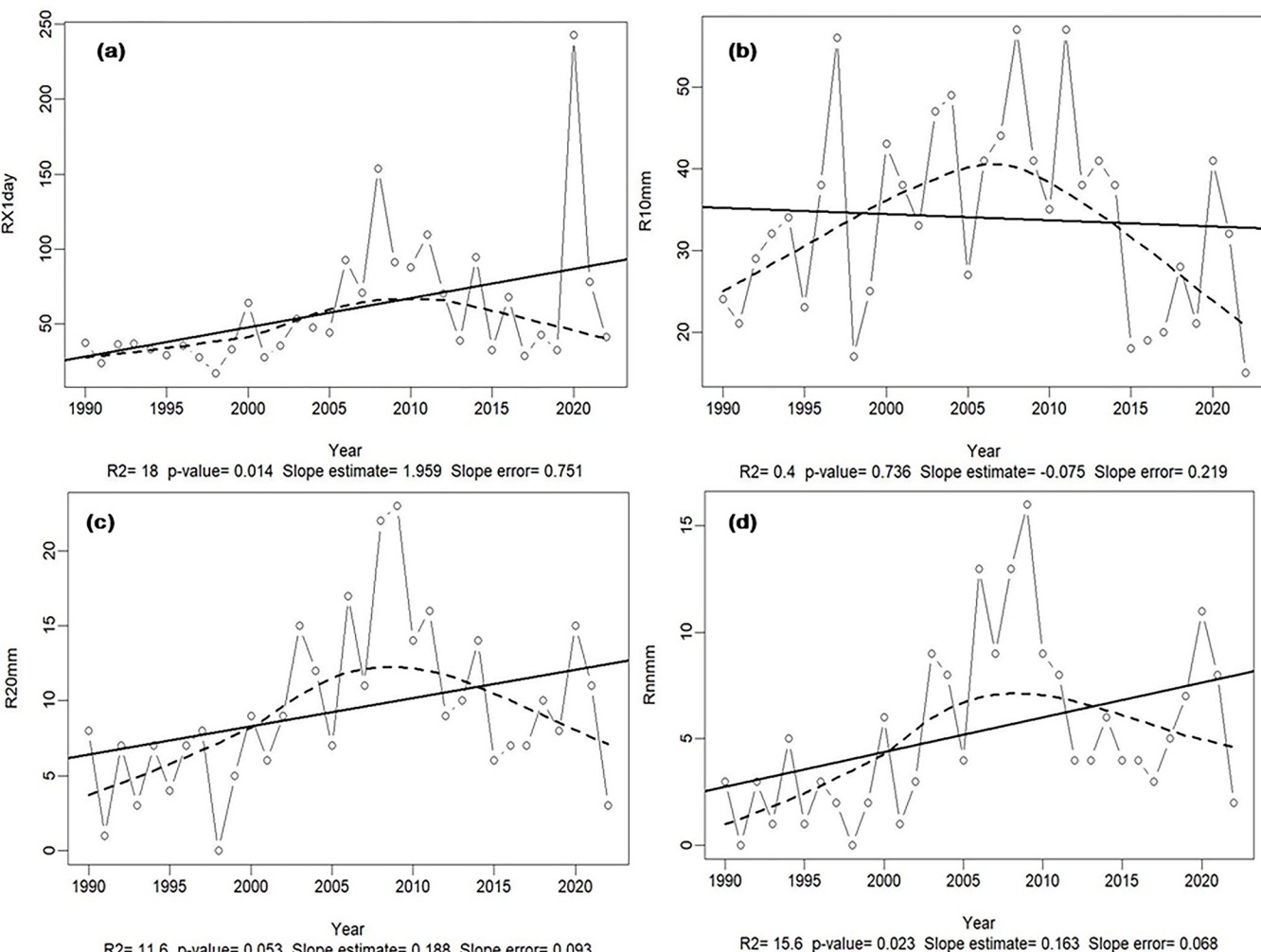

**Fig 1. Trends in climate data over the last three decades.** (**a**) Change in the number of rainy days, (**b**) intensity of rainfall >10 mm, (**c**) extreme rainfall events >20 mm, (**d**) extreme rainfall events exceeding 25 mm in the AEZ1 from 1990 to 2022. The inter-annual variation is given by the line graph with the small circles, the data series' trend by the dotted line, and the fitted linear trend by the solid line to detect whether a particular index is following an increasing or a decreasing trend over time. PRCPTOT: annual precipitation in mm, R10: The annual count of days with precipitation exceeding 10 mm, R20: The annual count of days with precipitation exceeding 20 mm, and RX5day: The maximum 5-day precipitation in a year. p<0.05: significant, p≥0.05 not significant, R2: coefficient of determination in %, names and significance of all the variables are described in **S1 Table**.

respectively. Analysis of the SPI (**S5 Fig**) for this study period shows slight variations of no more than ±0.5. In terms of monthly variations, two months (June and July) are or can be classified as "slightly dry". All other months remain "wet" to "very wet ". This is a typical feature of the equatorial climate (**Fig 2**).

In the AEZ 3, a humid tropical climate found in medium and high altitudes (1000 to 2500 m), there is strong increase in mean annual rainfall of up to 27% from 1990 to 2020 (**Fig 3A**). In contrast to the AEZ1, this zone experiences decreases in extreme precipitation (those of >20 and 25 mm) during the considered time (**Fig 3B**). Decreasing trend was also observed for daily rainfall events (**Fig 3C**). On the other hand, rainfall of less than 10 mm increased over the same period (**Fig 3D**). No significant changes were observed for minimum and maximum temperatures from 1990 to 2022. During the considered period, the lowest temperatures were between 14.8 and 15°C and average temperatures ranged from 17.8 to 18.8°C, i.e. an increase

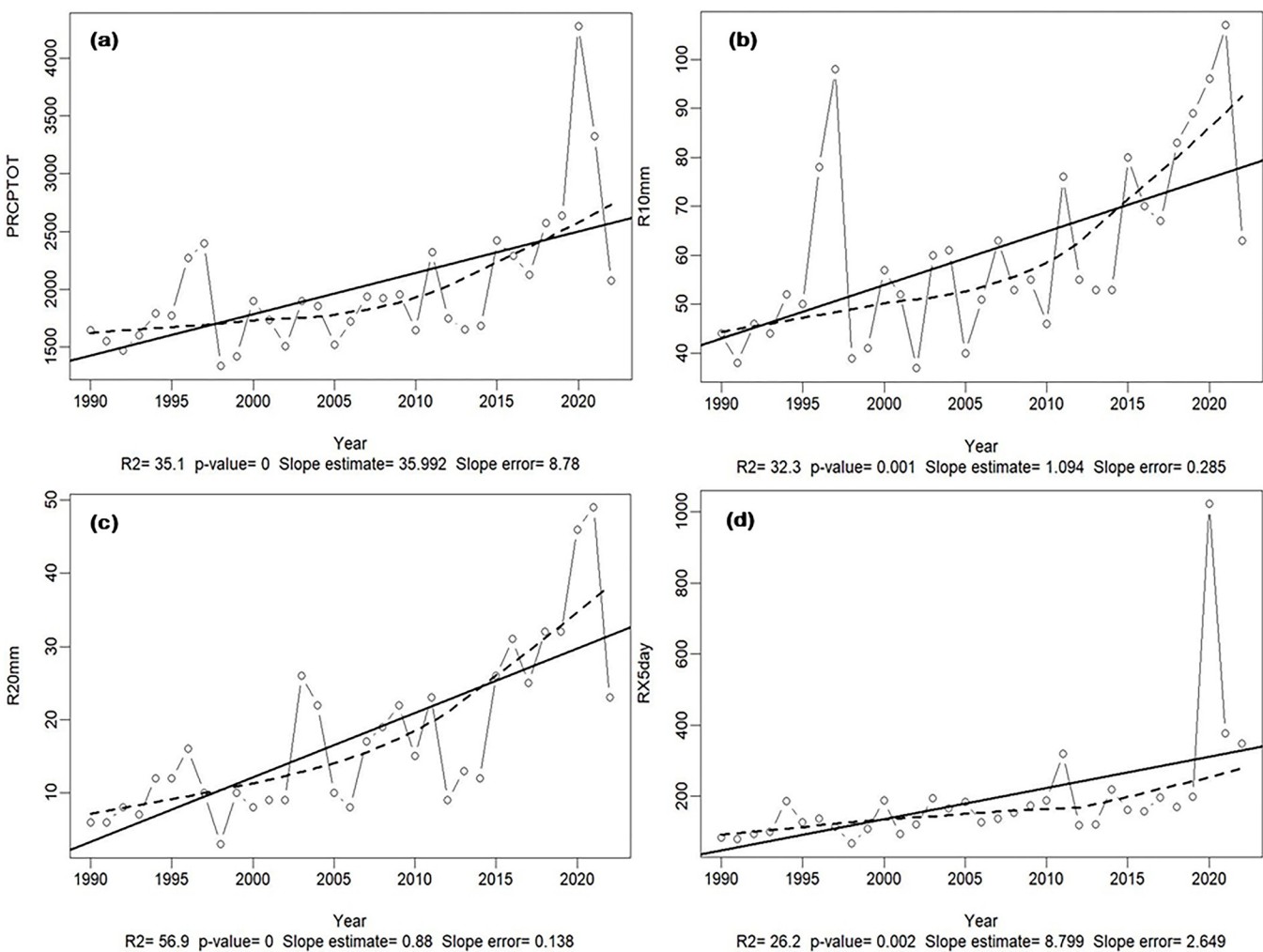

**Fig 2. Evolution of climate parameters in the AEZ2 from the 1990 to 2022 data series.** (**a**) Annual precipitation in mm (PRCPTOT), (**b**) The annual count of rainy days exceeding 10 mm (R10mm), (**c**) The annual count of rainy days exceeding 20 mm (R20mm), and (**d**) The maximum 5-day precipitation in a year (RX5day). The inter-annual variation is given by the line graph with the small circles, the data series' trend by the dotted line, and the fitted linear trend by the solid line to detect whether a particular index is following an increasing or a decreasing trend over time. $p<0.05$: significant, $p\geq0.05$ not significant, R2: coefficient of determination in %, names and significance of all the variables are described in **S1 Table**.

of 4.5% for the three decades. Maximum temperatures were ~27.2–27.8°C. SPI analysis predicts a succession of years that are wetter than dry, with only 3 months that could be classified as dry (June, July, and August). Three other months (September, October, and February), which are sometimes dry and sometimes wet, are subject to wide inter-annual variations.

The AEZ 4, in the highlands, had a similar trend to that of the AEZ 3, with strong variations in temperature and rainfall onset owing to the Foehn effect (**Fig 3**). This decrease in temperature impacts not only the water demand (i.e., the ETo) but also the plant growth rate, and thus extending the crop cycle.

## Estimated planting dates for yam cultivation across various South-Kivu AEZs

Based on focus group discussions with farmers on various planting dates across AEZs, five scenarios were formulated for planting dates: 15/09 (S1), 20/09 (S2), 30/09 (S3), 05/10 (S4), and

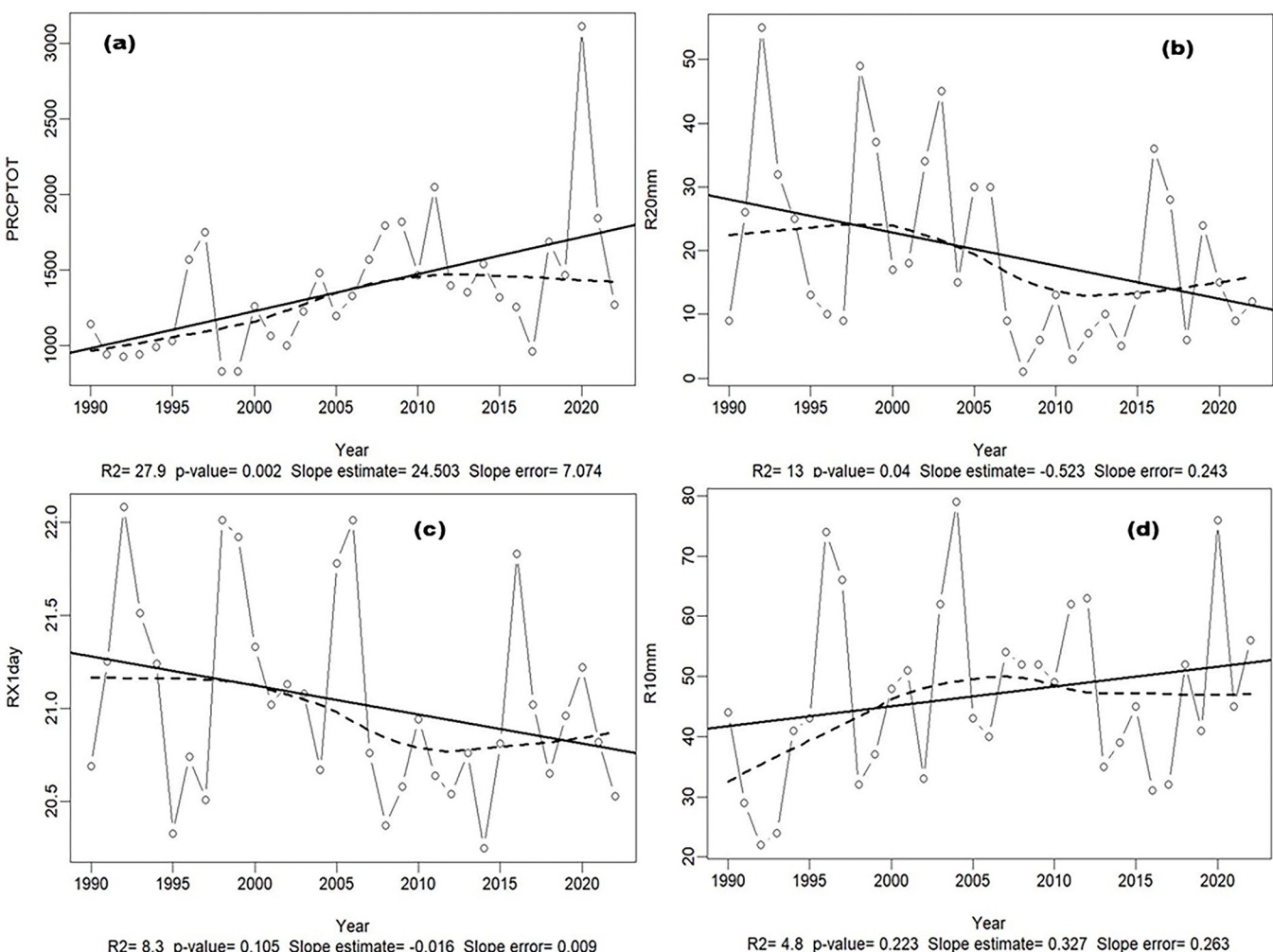

**Fig 3. Changes in climate data in the mid- to high-altitude humid tropical climate regions from 1990 to 2022. (a)** Total annual rainfall in mm (PRCPTOT), **(b)** number of rainy days exceeding 20 mm rainfall (R20mm), **(c)** The maximum 1-day precipitation in a year (RX1day), **(d)** number of rainy days exceeding 10 mm (R10mm). The inter-annual variation is given by the line graph with the small circles, the data series' trend by the dotted line, and the fitted linear trend by the solid line to detect whether a particular index is following an increasing or a decreasing trend over time. p<0.05: significant, p≥0.05: not significant, R2: coefficient of determination in %, names and significance of all the variables are described in **S1 Table**.

15/10 (S5). **Table 2** shows the estimated total crop water requirements, the actual rainfall, and the differences between them, considered here as the cumulative deficit that can be considered as the crop's irrigation requirement.

Based on the observed late onset of the rainy season in AEZ 1, AEZ 3, and AEZ 4, late planting scenario S5 (15/10) seemed the most promising, though it presents the highest risk of water deficit towards the tuber bulking and crop maturity phases. On the other hand, early planting scenarios S1 (15/09) and S2 (20/09) present the lowest water deficits but which occur at critical phases, such as sprouting and crop establishment. These scenarios fall in September, a month that is becoming increasingly dry with erratic rainfall (**Table 2**). It is, however, noteworthy that none of the scenarios could help meet the yam water requirements in the AEZ1, AEZ3, and AEZ4 since effective rainfall (Pmm) is inferior to the plant water demand (ETc), regardless of the planting scenarios. In such context, yam cultivation will require the use of soil water conservation practices for optimum plant growth and yield.

**Table 2. Overall yam water demand (ETc), actual (effective) rainfall, and total or cumulative water deficits by planting scenario across AEZs.**

| Test sites | AEZs | ETc mm/dec | Eff rain mm/dec | Irr. Req. mm/dec | Planting date scenario |
|---|---|---|---|---|---|
| Fizi | AEZ 1 | 1060.2a | 948.8a | 248.1b | 15/09 (S1) |
| | | 1059.5a | 944.3ab | 243.1c | 20/09 (S2) |
| | | 1057.9a | 939.9b | 247.2b | 30/09 (S3) |
| | | 1055.3b | 928.2c | 259.1a | 05/10 (S4) |
| | | 1054.6b | 921.2d | 266.8a | 15/10 (S5) |
| Walungu | AEZ 3 | 1123.3b | 1065.3c | 180.8a | 15/09 (S1) |
| | | 1127.8a | 1090.6a | 158.3c | 20/09 (S2) |
| | | 1127.4a | 1084.2b | 164.8b | 30/09 (S3) |
| | | 1126.1a | 1076.7b | 169.8b | 05/10 (S4) |
| | | 1129.4a | 1103.3a | 143.7c | 15/10 (S5) |
| Uvira | AEZ 1 | 1231.5ab | 1010.2a | 273.3c | 15/09 (S1) |
| | | 1229.2b | 999.4a | 284.7b | 20/09 (S2) |
| | | 1228.9b | 974.5b | 313.4ab | 30/09 (S3) |
| | | 1234.7a | 948.9c | 345.0a | 05/10 (S4) |
| | | 1239.4a | 931.0c | 357.7a | 15/10 (S5) |
| Kalehe | AEZ 2 | 1087.4a | 1124.8a | −110.9b | 15/09 (S1) |
| | | 1084.7a | 1112.1a | −122.4b | 20/09 (S2) |
| | | 1082.9a | 1098.8b | −132.6a | 30/09 (S3) |
| | | 1083.7a | 1105.8a | −129.3ab | 05/10 (S4) |
| | | 1082.5a | 1089.1b | −144.1c | 15/10 (S5) |
| Kabare | AEZ 3 | 1134.0a | 1103.3a | 143.9c | 15/09 (S1) |
| | | 1132.5a | 1090.6b | 158.3b | 20/09 (S2) |
| | | 1132.1a | 1084.2b | 164.7a | 30/09 (S3) |
| | | 1131.8a | 1076.7c | 170.4a | 05/10 (S4) |
| | | 1132.2a | 1065.3c | 182.8a | 15/10 (S5) |
| Mwenga | AEZ 4 | 943.7a | 1221.9c | −110.7a | 15/09 (S1) |
| | | 942.5a | 1203.1c | −121.7b | 20/09 (S2) |
| | | 942.2a | 1194.2b | −130.2b | 30/09 (S3) |
| | | 942.1a | 1183.4b | −132.1b | 05/10 (S4) |
| | | 942.6a | 1166.2a | −142.4c | 15/10 (S5) |
| Idjwi | AEZ 3 | 1112.3a | 1033.4a | 154.8b | 15/09 (S1) |
| | | 1111.0a | 1029.2a | 158.4b | 20/09 (S2) |
| | | 1108.7b | 1018.3b | 167.6b | 30/09 (S3) |
| | | 1107.9b | 1011.6c | 173.5ab | 05/10 (S4) |
| | | 1107.7b | 1000.5c | 185.0a | 15/10 (S5) |

AEZ: agroecological zone, ETc: cultural evapotranspiration, Eff rain: effective rainfall (in mm/dec), Irr. Req: irrigation requirement (in mm/dec), dec: decade, the green color represents the most favorable planting date for a particular AEZ, yellow is an intermediate (tolerable) planting date, while red color refers to less favorable planting date that should be avoided.

For the AEZ 2, planting can be conducted at any time in September and October (since rainfall exceeds the crop's water requirements). However, preferences could be inclined towards early planting scenarios S1 (15/09) and S2 (20/09), as they present low risk of flooding.

The analysis of water requirements per growth phase, corresponding to the cumulative water deficit, crop water requirements (ETc), and effective rainfall per growth phase, are presented in **S2 Table**. Of all the zones, the AEZ 2 and AEZ 4 had the lowest cumulative water

deficits per decade. It is noteworthy that critical phases for water supply are the first two crop growth stages (initial/sprouting and development/establishment). Results showed that all the proposed scenarios had very low deficits for the initial yam growth phase, regardless of the AEZ though deficits seemed greater in the AEZ 1 and AEZ 2. During the development phase, cumulative deficits are very low (less than 1 mm/dec). Plants require less water in the last growth phase, meaning that excess water should be avoided to prevent tuber rotting. Of the four AEZs, AEZ 1 and 4 had a high water accumulation at the last growth stage of the plant, which could have an impact on yield at harvest. In AEZs 2 and 3, on the other hand, the deficit is very low at 3.8 and 9.2 mm/dec, respectively, for the whole cycle. This implies that the cycle could be completed by the remnant soil water. **Fig 4** presents the combination of the proposed planting date scenarios and the four crop growth stages across the four agro-ecological zones.

The first two growth phases did not differ in terms of water requirements in the AEZ 2, regardless of the planting date scenarios (**Fig 4A**). However, the late planting scenario (S5) exhibited the highest water requirement at the third growth stage, with a decreasing demand towards the end of the cycle. For all other planting date scenarios, however, the water deficit

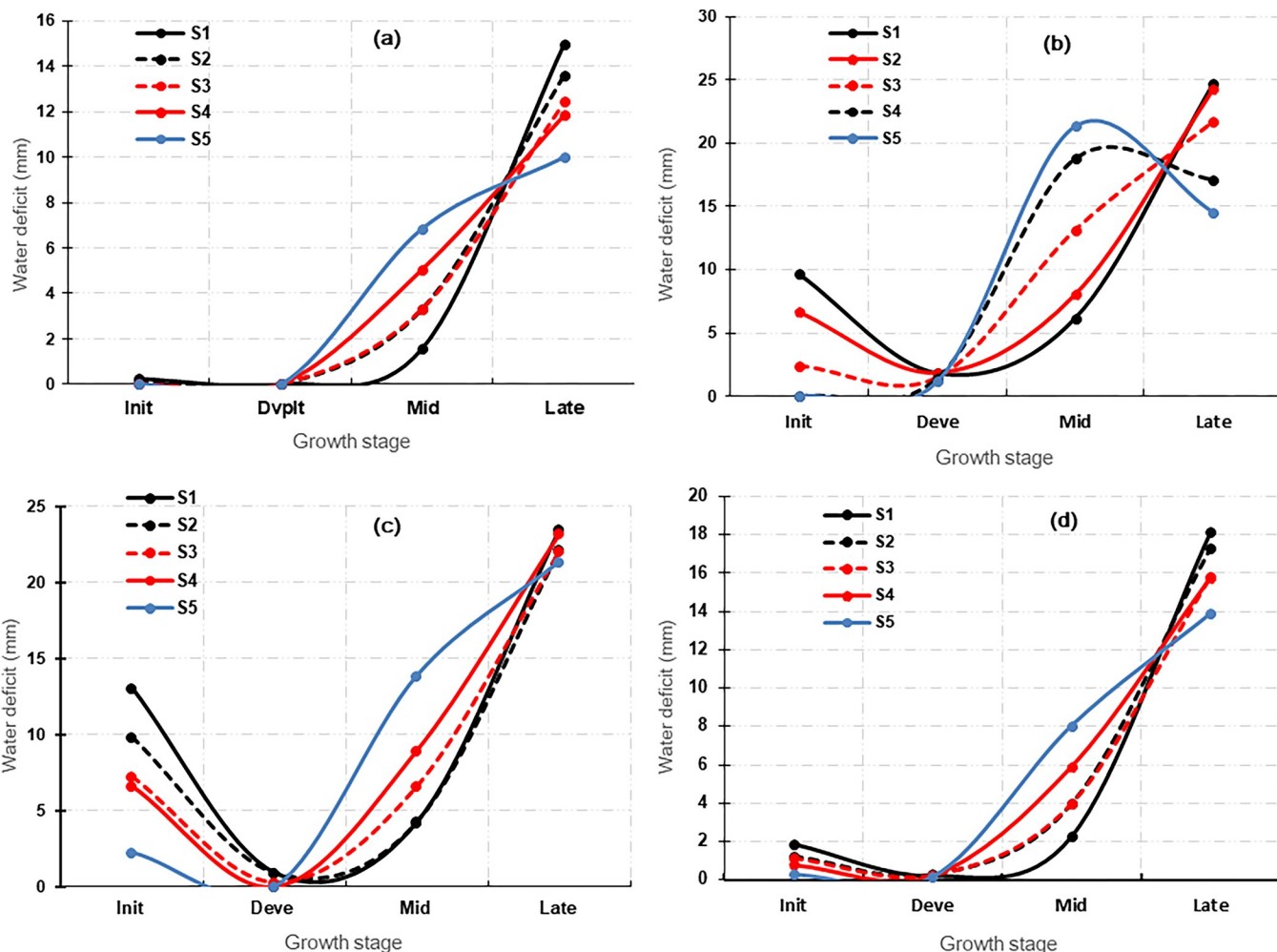

**Fig 4.** Variation in water demand (in mm) by development stage for five planting date scenarios proposed in (a) AEZ 2, (b) AEZ 4, (c) AEZ 1, and (d) AEZ 3. S1, S2, S3, S4, and S5 refer to planting scenarios. Init = initial phase, Deve = Dvlpt = development phase, Mid = intermediate (mid-) growth phase, and Late = maturity and senescence (late) phase.

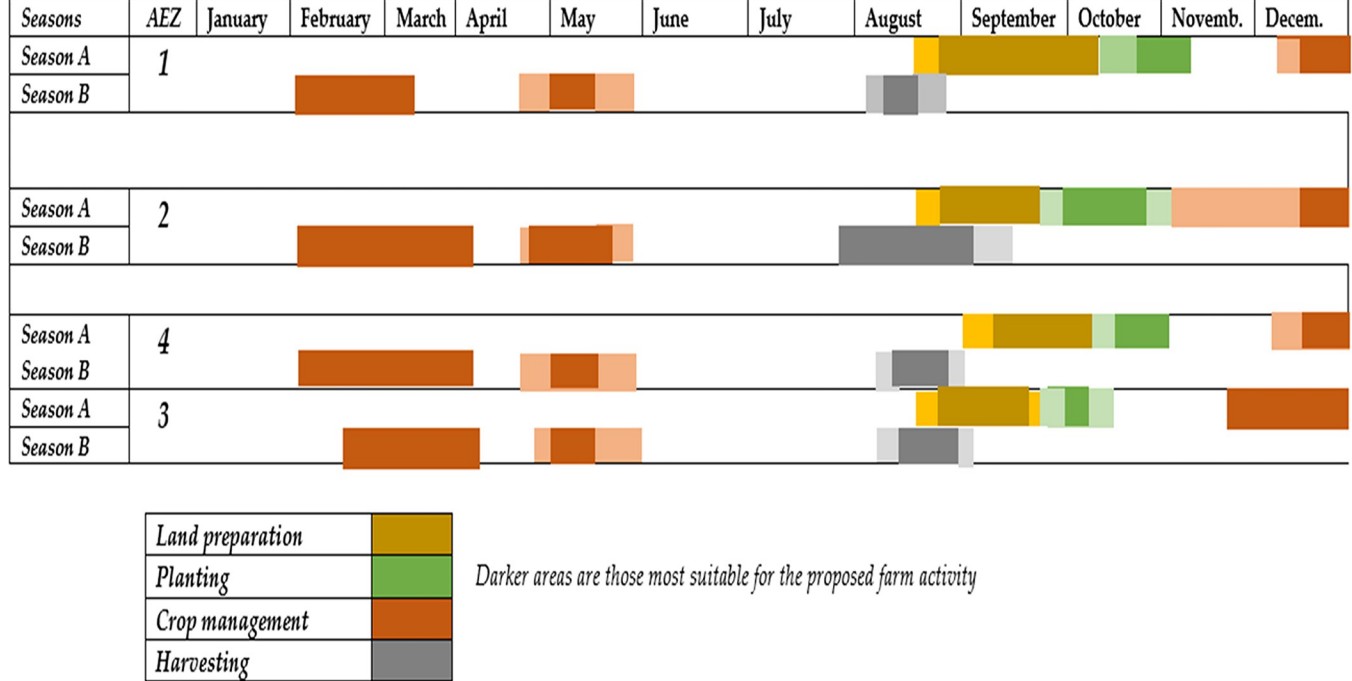

**Fig 5. Simplified yam crop calendar for various South-Kivu AEZs, eastern DRC.** Season A refers to the long rainy season (September to January) while Season B is the short rainy season (February to May). Crop management in yam includes weeding, manure application, staking, ridge maintenance, supplemental irrigation when applicable, etc. For a same color, the darker the color, the more suitable is the period for a given farm activity.

increased along the yam growth cycle. Early planting scenario S1 had minimal water deficit at this stage, and is therefore, recommended for this AEZ. On the other hand, the initial stages suffered from a significant deficit for AEZ 4 and AEZ 1 (**Fig 4B** and **4C**). However, deficits are lower for the late planting scenario (S5) as compared to early planting scenarios (S1 and S2) in both AEZs. In AEZ 3, the first three growth stages had low water deficits, that significantly increased at the last stage to favor tuber maturation.

For the AEZ 1, the crop water demand is often greater than the actual rainfall (**S6A** and **S6B Fig**). Planting is recommended between mid-October and mid-November when rainfall is abundant and regular, though mid-October is still characterized by erratic rainfall events, dry weeks alternating with torrential rains exceeding 20 and 25 mm a day. Though drought is much pronounced in early May onwards, there are water deficits observed in mid-January, February, and sometimes in March, calling for water management practices able to boost the soil water's useful reserves. It is noteworthy that soil conditions in the AEZ1 has low soil water retention capacity, as soils are mostly sandy with a high infiltration capacity [15]. It is, therefore, useful to recommend soil water conservation techniques that prolong soil water availability during dry periods.

The situation is significantly different in the AEZ 2 (**S7A** and **S7B Fig**). In this AEZ, only mid-June to mid-August show water deficits, which are also negligible. With heavy rains observed in this AEZ, the optimum planting date is the one with minimum water deficit's differences among growth phases, and which allows locating these water deficits at the maturity phase. Thus, optimum planting dates for the AEZ 2 are located in the first (S1) or second (S2) decade of September. Besides, it will be necessary to combine good planting dates with farming techniques that reduce excess water in the last growth stage to prevent tuber rotting.

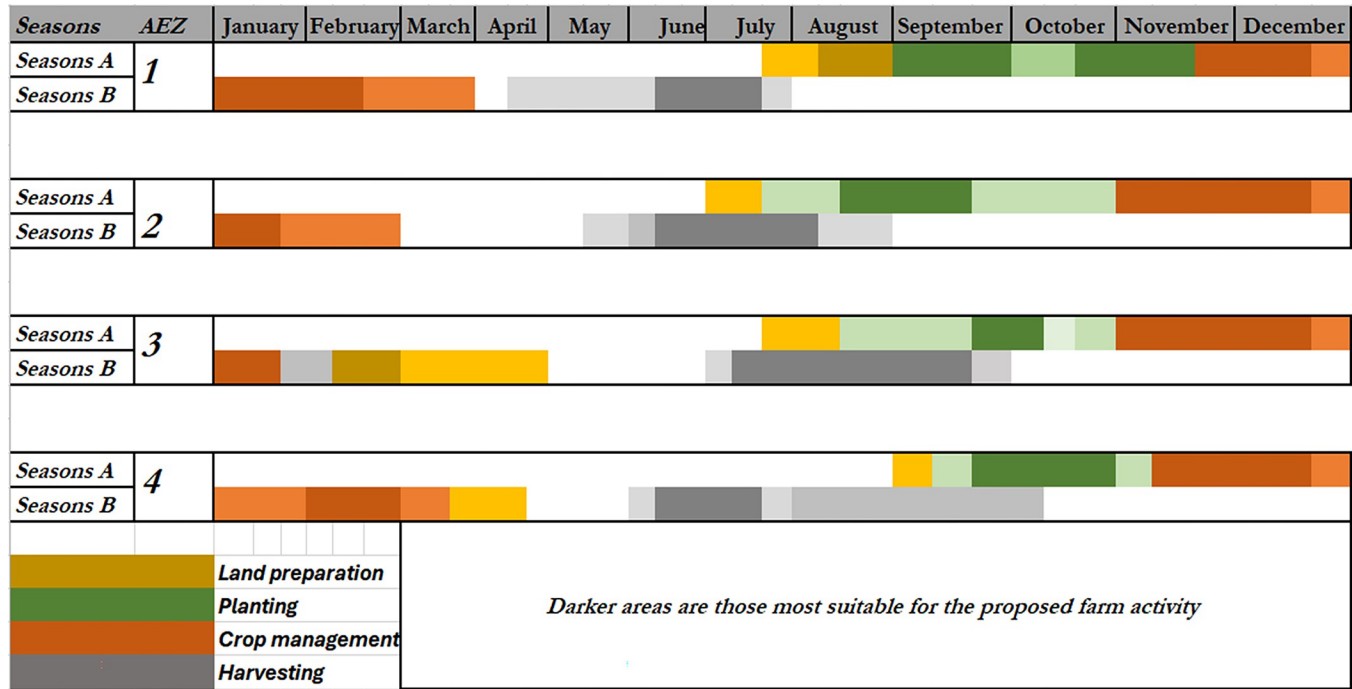

**Fig 6. Schematic presentation of the traditional yam crop calendar for various South-Kivu AEZs, eastern DRC.** Season A refers to the long rainy season (September to January) while Season B is the short rainy season (February to May). Crop management in yam includes weeding, manure application, staking, ridge maintenance, supplemental irrigation when applicable, etc. For a same color, the darker the color, the more frequent a period was referred to by local farmers for a given farm activity.

For the AEZ 3, the optimum planting date is between the second and third decades of October. Early planting scenarios present strong water deficits that could hamper crop sprouting or establishment (S8A, S8B Fig). Similarly, if the short rainy season (January and June) had to be explored, the water deficits observed in April would coincide with the active growth phase and thus hamper the crop's proper growth and development that could translated in reduced tuber yields. The high water demands observed in May, June, and July mean that this period should coincide with tuber maturation and crop senescence. After this period, the tubers can be harvested to allow land preparation for the next planting season.

The situation in the AEZ 4 is intermediate between AEZ 3 and AEZ 2 (S9A, S9B Fig). There are two main tipping points in the water cycle, between the first and the second decades of October, and the second decade of May. Two other tipping points occur from late January to mid-February. Although rains effectively return in early September in this AEZ, they do not meet the plant's water requirements. In fact, the first decade of September tends to be humid, then followed by a dry decade. This makes the decision on the optimal planting dates unpredictable. The best planting date scenario would be located between S3 and S4, i.e. from mid- to late October. Farmers practicing conservation tillage techniques in this AEZ can start planting in the preceding decade (i.e., S2), but with the risk of intensive crop management to control water deficit in late January, mid-February, and May. However, a risk is still foreseeable in this AEZ in view of rainfall disturbances and fluctuations. This AEZ is characterized by a multitude of microclimates owing to diversity of topographies and weather conditions that significantly fluctuate from one area to another, with major effects on rainfall distribution and intensity.

## Crop calendar for various South-Kivu AEZs

Combining all the results, we have suggested a crop calendar for each AEZ as presented in **Fig 5**. It suggests optimal planting dates from late October to early November, except for the AEZ 2 where the rain onset is much early. For all the AEZs, land preparation should be conducted from late August to early October, and should include field clearing, first and second ploughing, burial of organic matter and crop residues, and ridging. Weeding should be carried out in December, February or March, and in May. Gap filling and staking are to be coupled with the first weeding in December. The harvest will coincide with the dry season (from late July to early September).

With high climate variability observed in the AEZ 1 and AEZ 4, we hypothesized that shifting the planting season would allow minimize water deficits for yam. Thus, for these two AEZs, three other planting date scenarios were tested in the short rainy season. This season starts in mid-February and is locally referred to as season B. We then compared effective rainfall and ETc to determine the trend in water deficit for each decade and crop development stage. These simulations showed that planting yam in mid-February would lead to harvesting in December or early January. For such a scenario, crop water requirements rose to 1258 mm for an effective annual rainfall of only 940 mm, i.e., a water deficit of 424 mm. Besides, critical growth phases fell in unfavorable conditions: active growth phase with tuber initiation fell into the dry season (late April to early October) while the tuber maturation coincided with heavy rains that could rot mature tubers (**S10 Fig**). If the planting date is shifted to early March (05/03) and the harvest to late January, they would be a plant water requirement of 1248 mm for an effective rainfall of 933 mm, i.e., a cumulative water deficit of 405 mm. In general, all the tested off-season's planting date scenarios would exacerbate water deficits, and thus, they should be avoided.

While comparing the optimized crop calendar from the traditional calendar that is in current use (estimated using survey data reported in Mondo et al. [29]), we found that traditional calendar is less precise with a wide window for each activity as a proof of uncertainty and lack of consensus among farmers (**Fig 6**). This uncertainty is much pronounced in low altitude AEZ 1 severely affected by climate changes.

## Discussion

### Climatic risks associated with yam production across South-Kivu AEZs

Risk analyses revealed a disparity among AEZs, with several microclimates often found within an AEZ. Low-altitude zones were subdivided into two AEZs: those with low precipitation (AEZ1) and those with very high precipitation (AEZ2). In both AEZs, temperatures were high throughout the year. These AEZs are mainly located in the eastern territories, such as Uvira and the northern and north-eastern Fizi, and in the western and north-western territories, such as Mwenga, Shabunda and the north-western Kalehe. The AEZ 1 is characterized by high temperatures, low rainfall (the lowest in the province), and a more pronounced dry season that exceeds four months [15]. In fact, a year in this AEZ is subdivided into five dry months, five wet months, and two intermediate months. In such a dry tropical climate, SPI calculations showed series of five dry years followed by two to three moderately wet years. This significant climate change could be partly associated with the profound land use dynamics in recent decades in this AEZ, impacting local weather conditions [15, 64, 65]. On the other hand, the AEZ 2 has a transitional equatorial climate, with two slightly dry months (SPI ≤1), five wetter months (SPI>1.5), and three slightly wet months (SPI between 0.5 and 1). Compared to the baseline situation (1990), some traditionally wet months are now neutral (neither wet nor

dry). It is, however, noteworthy that there is no significant water deficit during the two dry months, since soil water reserves are still enough to sustain crops. Since yam is sensitive to flooding, excess rains observed in these areas, especially during tuber maturation, could cause tuber rotting if planting dates or varieties are not properly decided [28, 66, 67].

Like the low-altitude dry tropical climate zone (AEZ1), the mid-altitude humid tropical climate zone (AEZ3) experiences high climate variability. For instance, the number of rainy days has decreased, there is a significant rise in extreme rainfall events (pmm > 20–25 mm), as well as rain episodes exceeding five days, resulting in rapid soil losses through water erosion and crop losses through silting [68]. These results agree with several other reports from the region showing inter-annual rainfall variability in the mid-altitude humid tropical climate zones of Walungu and Kabare [69]. These authors showed that series of three to four wet years are repeatedly followed by one or two slightly to moderately dry years (SPI = -0.5 to -1.2). Besides, there is a prolonged dry season extending from June to the first week of October, meaning that September and October that were traditionally wet had shifted to dry [69]. Thus, the existing crop calendar that recommends mid-September as the optimum planting season is no longer viable and should be shifted to fit current climate realities. Though some microclimates within the AEZ3 could still apply the existing crop calendar owing to the weather regulating actions of the Lake Kivu and the Kahuzi Biega National Park [70], other regions falling under the AEZ3 should quickly shift date to adapt with climate variability in South-Kivu.

Many climate change signals or indicators as perceived by local communities across South-Kivu AEZs have also been ascertained by this study. In general, the climate change in South-Kivu has been linked with the occurrence and or increase in extreme weather events, rising temperatures, and rainfall (or their decrease in some AEZs), but above all, to the late rainfall onset [26, 69]. At some AEZs, communities listed also an increase in the frequency of extreme rainfall events characterized by high intensity rains that could last more than five days [69]. These indigenous indicators have been ascertained by the 1990–2022 data series. In fact, in almost all zones (except the AEZ 3), increases in rainfall events exceeding 20 and 25 mm, in the number of rainy days (except the AEZ 1), and in the cumulative annual rainfall amounts were recorded from 1990 to 2022. Although all AEZs receive annual rainfalls exceeding the total yam water demand (i.e. >1100 mm), rainfall distribution is a major challenge. Such situation was reported by several other studies in South-Kivu [15, 26, 69, 71, 72]. Based on this study and a previous report on yam land suitability in South-Kivu [29], the AEZ4 is a humid tropical zone with very high altitudes. In addition to soil depletion associated overexploitation (the region being densely populated) and soil erosion, this AEZ is characterized by low soil organic matter, affecting the soil ability to retain rainwater. This implies that soils (dominated by clayey Ferralsols) from this AEZ are likely to suffer from water deficit in case of prolonged dry spells and erratic rainfall, calling for the need of promoting soil water conservation practices [68]. Fortunately, areas characterized by AEZ4 are located between Lake Kivu and the Kahuzi Biega National Park that are believed to regulate climatic conditions in the area to the extent that averages of climatic parameters such as rainfall and temperatures have remained unchanged over time [70]. A yam crop in the field settings in South-Kivu highlands, eastern DRC, is presented in **S11 Fig**.

## Adjustments of the yam crop calendar and other measures to adapt to climate hazards

There is an urgent need for actions to adapt to climate changes in South-Kivu by encouraging local level adaptation efforts to strengthen yam farmers' capacity to deal with the adverse effects of climate change. Climate change adaptation requires a combination of adaptive

strategies, informed decision-making, and supportive policies to enhance resilience and ensure food security. Therefore, adjusting the crop calendar is not sufficient for climate change adaptation, four main other elements can be recommended as discussed next. Firstly, early/delayed planting and crop selection. Here, farmer can adjust planting dates based on coming climate patterns while choosing crops and or varieties that are more resilient to changing climate conditions, such as drought-tolerant or heat-resistant varieties or crops [67]. As also reported by Iseki et al. [67], this study do not recommend shifting the planting date backward since it may result in significant yield reduction. Overall, crop diversification can also help mitigate risks associated with climate variability [4, 33]. Secondly, combining both efficient water management practices and integrated pest management (IPM) practices. These comprise rainwater harvesting, irrigation scheduling, combined with control pests and diseases that may emerge under changing climate conditions [15, 73, 74]. As the region suffer from land degradation through erosion, adoption of agroforestry and soil conservation can be integrated [68]. Planting trees and integrating them with crops can help improve soil fertility, water retention, and biodiversity, making agricultural systems more resilient to climate change [75]. This practice would be particularly useful for yam farmers as trees would provide living stakes for yam. Thirdly, it would be crucial to provide farmers with timely and accurate information about weather forecasts, climate projections, and best agricultural practices to empower them to make informed decisions and to adapt to changing conditions more effectively. Lastly, governments and policymakers should support adaptation efforts by implementing policies that incentivize sustainable agricultural practices, invest in climate-resilient infrastructure, and provide financial assistance and insurance schemes for farmers affected by climate-related disasters.

The above-mentioned adaptation measures include not only adjusting crop calendar, but also building on the traditional knowledge systems to devise or introduce technologies that suit the local conditions [4]. In the case of South-Kivu, actions should be oriented towards agronomic practices that improve soil water reserves to cope with dry spells and erratic rainfall. These practices include, for example, the use of rainwater harvesting techniques such as tied ridging, half-moon, and zaï pits, combined with organic matter and micro-irrigation [4, 15, 76]. These techniques should be complemented with a judicious choice of planting date and harvest or the use of resilient crop varieties, mainly early maturing ones. These techniques had been successfully tested by Bagula et al. [15, 76, 77] in the South-Kivu's AEZ 1, as well as other scholars in neighboring countries and regions [78–80].

It is noteworthy that this article has not included future trends in climate data analyses due to limited documentation on climate models (global, regional, or local) adapted to the South-Kivu province, making difficult to suggest future adjustments to the proposed crop calendar. However, based on past trends and conclusions by Bagula et al. [15, 72] for the AEZ1 (Ruzizi plain and Fizi territory), fluctuations in rainfall and temperature, combined with low communities' adaptation capacity, will significantly worsen local communities' vulnerability. It is, therefore, urgent that decision-makers propose significant alternatives for mitigating and attenuating climate hazards in the area. This implies adjusting the crop calendar based on this study's recommendations to fit current climate realities for each AEZ to maximize crop yields and reduce climate shocks. However, before such calendar adjustments, the proposed calendar should be tested across different microclimates, using participatory research approach, to ensure risks are minimized and high adoption as farmers' feedback will be considered in refining agricultural activities' planning. This calendar will reduce agriculture's vulnerability among yam farmers as it considers both scientific and farmers' local knowledge instead of relying on a calendar that has been based solely on indigenous knowledge and that has become obsolete with climate change. As stated above, further studies are needed for long-term

calendar adjustment by assessing future climate change (e.g. 2040s, 2050s, 2080s and end of the century) and yam land suitability in the future when rainfall and temperature will likely be very different from the changes noted between 1990 and 2022. This implies also long-term monitoring and evaluation studies to assess the effectiveness of crop calendar optimization strategies in yam cultivation under changing climatic conditions. Other key areas that could be examined in future studies are: (a) the future impact of climate change on the crop yield as water demand changes and (b) the impact of pest as future water availability (including drought) could become more challenging. Integrating stakeholder perspectives through engagement with local farmers, agricultural extension workers, and other stakeholders to better understand their knowledge, perceptions, and needs regarding climate change adaptation in yam cultivation, and incorporating their perspectives into future research and interventions. Exploring the integration of climate-smart agricultural practices, such as agroforestry, soil conservation, and water management techniques, into yam cultivation systems to enhance resilience and sustainability in the face of climate change is another option to deeply analyze in the future.

## Conclusions

This study sought to reduce yam farming vulnerability to climate change by developing a crop calendar adapted to current climate change realities for various AEZs in South-Kivu, eastern DRC. Local knowledge and simulation using CROPWAT 8.0 tool were combined to provide a robust crop calendar for yam cultivation. We found significant climate variability across all South-Kivu's AEZs, translated by drought or flooding depending on AEZs. The low-altitude AEZ 1 recorded the highest water deficit, as the atmospheric demand largely exceeded rainfall amounts. It experienced also high variability in rainfall distributions, making the traditional yam crop calendar irrelevant. On the other hand, low altitude transitional equatorial zone (AEZ 2) had the least water deficit, providing a flexibility in farming activities' planning. Besides, the proposed calendar is highly reliable in this AEZ. Intermediate climate change effects are observed for the AEZ 3 and AEZ 4 located at high altitudes, though they experience high inter-annual variability especially for rainfall, except in the vicinities of the Lake Kivu and the Kahuzi Biega National Park where these two ecosystems regulate local climate. Regardless of the AEZ, late planting is encouraged to cope with late rainfall onsets in most AEZs. However, shift in planting date is not enough to cope with dry spells and erratic rainfall from the climate change and should be complemented by appropriate farming practices such as the use of tolerant varieties, irrigation, rainwater harvesting techniques, adequate weather forecasts, and policies incentivizing climate-smart agriculture promotion.

Simulations following future climate change scenarios as proposed by the IPCC Assessment Report (AR) VI (2022) [81] are necessary to propose future changes that might be integrated to the yam crop calendar for long-term reliability. Further efforts are necessary to design varieties tolerant to climate change and adapted to South-Kivu's agroecological and farming conditions. Our research contributes to the growing body of knowledge on climate change adaptation in agriculture and specifically for promotion of climate-smart yam farming system in South-Kivu, and provides practical recommendations for policymakers, extension workers, and yam farmers to enhance resilience and food security in the region. By working together and embracing innovative solutions, we can effectively address the challenges posed by climate change and ensure a more resilient and sustainable future for yam cultivation in South-Kivu. Such a study thus serves as the starting point for the establishment of a program aimed at valorizing neglected and underutilized crops such as yam in the region.

## Supporting information

**S1 Fig. Different windows of CROPWAT 8.0 used for parametrization and assessment of the water demand of yam in different agroecological zones in South-Kivu (simulated values are presented in yellow column with the input values).**
(DOCX)

**S2 Fig. Diagram of the water balance elements used in the soil parameterization of the CROPWAT tool.**
(DOCX)

**S3 Fig. Umbrothermal diagrams of South-Kivu AEZs.**
(DOCX)

**S4 Fig.** Calculation of SPI for dry tropical climate (AEZ1) for each year (a) and each month (b).
(DOCX)

**S5 Fig.** Calculation of SPI for equatorial (AEZ2) and humid tropical climates temperate with altitude (AEZ4) for each year (a) and each month (b).
(DOCX)

**S6 Fig.** Distribution of effective rainfall per month (a) and decadal evaporative demands (b) of the yam crop in the AEZ 1.
(DOCX)

**S7 Fig.** Distribution of crop water demand (ETc) and effective rainfall per month (a) and decade (b) throughout the year in the AEZ 2.
(DOCX)

**S8 Fig.** Distribution of effective rainfall and crop water demand by month (a) and decade (b) in the AEZ 3.
(DOCX)

**S9 Fig.** Distribution of plant water demand (ETc) and effective rainfall per month (a) and decade (b) of the year for the AEZ 4.
(DOCX)

**S10 Fig.** Distribution of plant water demand (ETc) and effective rainfall by month (a) and decade (b) of the year.
(DOCX)

**S11 Fig. Pictures of the yam crop in the field settings in South-Kivu, eastern DRC.**
(DOCX)

**S1 Table. Indices names, definition and units used for each indicators.**
(DOCX)

**S2 Table. Estimated crop water requirement (ETc), effective rainfall (Eff rain), and the cumulative water deficit (in mm/dec) for yam across South-Kivu AEZs.**
(DOCX)

## Acknowledgments

Authors are thankful to farmers who participated in focus group discussions to select test planting date scenarios as well as weather stations that availed field data used for bias corrections.

## Author Contributions

**Conceptualization:** Jean M. Mondo, Géant B. Chuma, Katcho Karume, Anthony Egeru.

**Data curation:** Jean M. Mondo, Géant B. Chuma, Henri M. Matiti, Jacques B. Kihye, Espoir M. Bagula.

**Formal analysis:** Jean M. Mondo, Géant B. Chuma, Henri M. Matiti, Jacques B. Kihye, Espoir M. Bagula.

**Funding acquisition:** Anthony Egeru, Jackson-Gilbert M. Majaliwa, Paterne A. Agre, Patrick A. Adebola, Asrat Asfaw.

**Methodology:** Jean M. Mondo, Géant B. Chuma, Espoir M. Bagula, Charles Kahindo, Anthony Egeru, Jackson-Gilbert M. Majaliwa, Asrat Asfaw.

**Project administration:** Paterne A. Agre, Asrat Asfaw.

**Resources:** Jean M. Mondo, Patrick A. Adebola.

**Software:** Géant B. Chuma.

**Supervision:** Katcho Karume, Charles Kahindo, Anthony Egeru, Jackson-Gilbert M. Majaliwa, Paterne A. Agre, Patrick A. Adebola, Asrat Asfaw.

**Validation:** Jean M. Mondo, Géant B. Chuma.

**Visualization:** Géant B. Chuma.

**Writing – original draft:** Jean M. Mondo, Géant B. Chuma, Jacques B. Kihye, Katcho Karume, Charles Kahindo, Anthony Egeru, Jackson-Gilbert M. Majaliwa, Paterne A. Agre, Patrick A. Adebola, Asrat Asfaw.

**Writing – review & editing:** Jean M. Mondo, Géant B. Chuma, Espoir M. Bagula, Paterne A. Agre, Patrick A. Adebola, Asrat Asfaw.

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
