## [Decision Letter · Decision Letter 0]

25 Jun 2024

PONE-D-24-20765

Crop calendar optimization for climate change adaptation in yam farming in South-Kivu, eastern D.R. Congo

PLOS ONE

Dear Dr. Chuma,

Thank you for submitting your manuscript to PLOS ONE. After careful consideration, we feel that it has merit but does not fully meet PLOS ONE’s publication criteria as it currently stands. Therefore, we invite you to submit a revised version of the manuscript that addresses the points raised during the review process.

The work presented is very important, but few minor concerns are noted below:

Please provide further justification of why CROPWAT was used over another model such as AquaCrop which can do the same type of analysis. You also mentioned that you have used NASA power data and corrected it using the local weather data. Kindly mention the RMSE if you have done. Directly depending upon Power data for assessing daily weather conditions, may not be correct especially when you are using it for further applications like optimizing crop calendars

There is no presentation of the traditional calendar that is in current use; this is a major omission and should be included. More details could be provided in the introduction (lines 108-116) and even a hypothetical/estimated graphic of the traditional calendar would be useful. An analysis to show how the new calendar improves on the old one would also be useful.

The discussion is somewhat repetitive but does not include what will happen in the future e.g. 2040s, 2050s, 2080s and end of the century, when rainfall will likely be very different from the changes noted between 1990 and 2022.

There is no key explaining the plot of the charts. For example, in figures 2-4, the authors could explain what each graph represents. Please provide a key for each panel of graphs. A key similar to that given in figure 5 would be useful.

We look forward to receiving your revised manuscript.

Kind regards,

Angela T. Alleyne, Ph.D

Academic Editor

PLOS ONE

Journal Requirements:

4. Please note that funding information should not appear in any section or other areas of your manuscript. We will only publish funding information present in the Funding Statement section of the online submission form. Please remove any funding-related text from the manuscript.

5. We note that your Data Availability Statement is currently as follows: All relevant data are within the manuscript and its Supporting Information files.

6. We note that Figure 1 in your submission contain map/satellite images which may be copyrighted. All PLOS content is published under the Creative Commons Attribution License (CC BY 4.0), which means that the manuscript, images, and Supporting Information files will be freely available online, and any third party is permitted to access, download, copy, distribute, and use these materials in any way, even commercially, with proper attribution. For these reasons, we cannot publish previously copyrighted maps or satellite images created using proprietary data, such as Google software (Google Maps, Street View, and Earth). For more information, see our copyright guidelines: http://journals.plos.org/plosone/s/licenses-and-copyright.

Reviewers' comments:

Reviewer's Responses to Questions

**Comments to the Author**

1. Is the manuscript technically sound, and do the data support the conclusions?

Reviewer #1: Yes

Reviewer #2: Yes

2. Has the statistical analysis been performed appropriately and rigorously? 

Reviewer #1: Yes

Reviewer #2: Yes

3. Have the authors made all data underlying the findings in their manuscript fully available?

Reviewer #1: Yes

Reviewer #2: Yes

4. Is the manuscript presented in an intelligible fashion and written in standard English?

Reviewer #1: Yes

Reviewer #2: Yes

5. Review Comments to the Author

Reviewer #1: 1.Is the Manuscript technically sound and of the data support the conclusions?

• Overall authors try to present sound basis for paper, but some gaps exist. The focus of the study is too narrow to make it useful and applicable beyond the study area.

o Efforts should be made to make comparisons with other areas in the DR, in Africa and other developing states

o Some focus/ comparison should be made with other root crops and perhaps provide a contrast with an above ground crop.

• Model Justification: Clearer justification of why CROPWAT was used over another model. For example, AquaCrop have been used to do the very same analysis.

• Traditional calendar: There is no presentation of the traditional calendar that is in current use; this is a major omission and should be included. More details could be provided in the introduction (lines 108-116) and even a hypothetical/estimated graphic of the traditional calendar would be useful. An analysis to show how the new calendar improves on the old one would also be useful.

o A comparison of the calendars could be presented as a figure 6b, or figure 7.

o Provide more details on calendar use (for any other crop) in Africa or the DRC.

• The conclusion does not follow logically from the analysis. The authors claim that they sought to reduce yam vulnerability to climate change by developing a crop calendar, but this intervention only addresses rainfall. There is no discussion on what will happen in the future e.g. 2040s, 2050s, 2080s and end of century, when rainfall will likely be very different from the changes noted between 1990 and 2022. More due to other environmental changes under climate change, future water demand will change.

o For example studies have shown that because atmospheric CO2 will increase, some crops may not need to open their stomata as long (per unit time) so water loss (and hence crop water demand) could be less.

o The authors noted the exclusion of future period analysis (lines 623-630), but there are models and studies that have been done and can be consulted via the latest IPCC reports for the African region.

• Other key areas that could be examined are:

o Impact on crop yield as water demand changes;

o Impact of pest as future water availability (including drought) could become more challenging;

2.Has the statistical analysis been performed appropriately and rigorously

• Further work is needed on the analysis: The authors have made an attempt to present data to support their claim, but more work is needed for it to make a compelling case.

o Climate trends: It would be useful to do an assessment of changes in climate over a longer period perhaps 50 years (1970-2020) and also to compare the changes in prior decades with the one in question. For example an earlier baseline period could be used (say 1961-1990) and changes before and after this be compared to see the magnitude of the changes and whether there is a continued linear trend.

Using additional graphs to depict changes of other decades would be useful.

It would also be useful to compare the changes noted in the study region with climate of other African countries and other areas in the region. Is it a typical or expected change? Does the region represent a microclimate? Limitations: as the data used is not actual station data (but grided data corrected by station data), the authors should explain the limitations of this approach, including the effect of distance of the station from the study area, the number of stations used, challenges with bias correction.

o Temperature and rainfall are distinctly different variables. One is stochastic (rainfall) while the other is continuous (temperature). For this reason, it is more useful to represent rainfall as relative (percentage change), but temperature absolute change (e.g. 0.5°C increase or decrease). For all intent and purposes a 12%increase in rainfall is not comparable to the same change in temperature since the quantities and baselines are demonstrably different.

• Charts: There is no key explaining the plot of the charts. For example, in figures 2-4, the authors could explain what each graph represent. The user is left wondering what each plot in the series represent. One intuitive view would be to interpret that the interannual variation is given by the line graph with the small circles, the mean by the dotted line and the trend by the solid line. Or is it something else? Please provide a key for each panel of graphs. A key similar to that given in figure 5 would be useful.

3. Have the authors made all the underlying data available…?

Yes… but as noted in 2 above additional data analyses are needed.

• Suggest using a different colour gradient (maybe different colours) for table 2. This would make the difference (high, medium and low) favourable planting dates more readily discernible.

4. Data presented in intelligible fashion, written in standard English

Yes

Specific comments:

Lines 46 and 60: Decide whether the term “rainfed” will be hyphenated or not and be consistent.

Line 155: Fig 1. Explain the descriptions in brackets, ie, low, medium etc.

Line 205 onward: What was the yield response factor? What was the critical depletion fraction (depletion level) used?

Line 211: is “P” monthly precipitation? If so, state that.

Lines 229-233: The relevance in that section is not seen. Could be placed in the Introduction or section on Method under Climate.

Line 255 Table 1- there is a typo. Depletion is currently “deplection”.

Add acronym TAW to table 1.

Line 330-335- What is the range of the annual rainfall?

Lines 337-345- Present the range of annual rainfall with minimum and maximum temperatures.

Line 442: Check Table number.

Line 432: Not seeing AEZ 4 in Table 2. Was a reason presented?

Line 519: Check grammar.

Would have liked to see a discussion of the relationships between the soil properties and the moisture levels especially in AEZ 4.

Reviewer #2: The authors have conducted a very relevant research, optimizing the crop calendar for Yam crop, appreciating the re effort, however, there are some minor errors which needs to be corrected.

1. In the Abstract : Use bias correction was carried out, instead of Edited and Corrected using local weather stations’ records.

2. droughts coinciding with yam critical growth phases, instead use coinciding with critical growth phases of yam , there are such grammatical errors which needs attention through out the manuscript. Kindly rectify them.

4. You have mentioned that you have used NASA power data and corrected it using the local weather data. Kindly mention the RMSE if you have done. Directly depending upon Power data for assessing daily weather conditions , may not be correct especially when you are using it for further applications like optimizing crop calendars

3. In fig. 5, mention unit of measurement in the Y axis value

4. Fig. 6. . In the Simplified yam crop calendar Kindly mention in detail, the term ‘ crop management’ in detail, it includes Planting and harvesting too, more over there are other color codes in the fig, which are not mentioned in the legend, Kindly add them in the legend

5. In the discussion section….the word ‘report’ can be replaced by article … It is noteworthy that this report has not included

6. Discussion is lengthy, with repetitive contents of results, that can be removed. Such as ….

An in-depth assessment of the climate risks associated with each planting date scenario was 573 carried out. This included an analysis of rainfall, temperatures, and other climate factors bearing influence on successful yam cultivation……

7. Appreciating the authors to add one or more pictures of the YAm crop in the field settings in the annexures/ supplementary files

6. PLOS authors have the option to publish the peer review history of their article (what does this mean?). If published, this will include your full peer review and any attached files.

Reviewer #1: No

Reviewer #2: **Yes: **Dhanya Punnoli

---

## [Author Response · Author response to Decision Letter 0]

14 Aug 2024

All the responses were added in the uploarded "RESPONSE TO REVIEWER COMMENTS

---

## [Editor Report · Decision Letter 1]

20 Aug 2024

Crop calendar optimization for climate change adaptation in yam farming in South-Kivu, eastern D.R. Congo

PONE-D-24-20765R1

Dear Dr. Chuma,

We’re pleased to inform you that your manuscript has been judged scientifically suitable for publication and will be formally accepted for publication once it meets all outstanding technical requirements.

Kind regards,

Angela T. Alleyne, Ph.D

Academic Editor

PLOS ONE
---

## [Editor Report · Acceptance letter]

22 Aug 2024

PONE-D-24-20765R1 

PLOS ONE

Dear Dr. Chuma, 

I'm pleased to inform you that your manuscript has been deemed suitable for publication in PLOS ONE. Congratulations! Your manuscript is now being handed over to our production team.

Kind regards, 

on behalf of

Dr. Angela T. Alleyne 

Academic Editor

PLOS ONE